# Effect of Lignin Content on Properties of Flexible Transparent Poplar Veneer Fabricated by Impregnation with Epoxy Resin

**DOI:** 10.3390/polym12112602

**Published:** 2020-11-05

**Authors:** Mengting Lu, Wen He, Ze Li, Han Qiang, Jizhou Cao, Feiyu Guo, Rui Wang, Zhihao Guo

**Affiliations:** 1College of Materials Science and Engineering, Nanjing Forestry University, Nanjing 210037, China; lumengting12345612@163.com (M.L.); lze0125@163.com (Z.L.); qianghan1016@163.com (H.Q.); cjz6127@163.com (J.C.); gfy1558@163.com (F.G.); ruiwang0909@163.com (R.W.); gzh3398@163.com (Z.G.); 2Innovation Center of Efficient Processing and Utilization of Forest Resources, Nanjing Foresry University, Nanjing 210037, China

**Keywords:** fast-growing polar, flexible transparent poplar veneer, optical properties, tensile strength, thermal stability

## Abstract

In this work, poplar veneer (PV) rotary-cut from fast-growing polar was delignified to prepare flexible transparent poplar veneer (TPV). Lignin was gradually removed from the PV and then epoxy resin filled into the delignified PV. The study mainly concerns the effect of lignin content on microstructure, light transmittance, haze, tensile strength, and thermal stability of the PVs impregnated with epoxy resin. The results indicate that the lignin could be removed completely from the PV when the delignification time was around 8 h, which was proved by FTIR spectra and chemical component detection. Moreover, according to SEM observation and XRD testing, the porosity and crystallinity of the PVs were gradually increased with the removal of lignin. Also, the optical properties measurement indicated that the light transmittance and haze of the TPVs gradually increased, and the thermal stability also became more stable as shown by thermogravimetric analysis (TG). However, the tensile strength of the TPVs declined due to the removal of lignin. Among them, TPV_8_ exhibited excellent optical properties, thermal stability, and tensile strength. Consequently, it has great potential to be used as a substrate in photovoltaics, solar cells, smart windows, etc.

## 1. Introduction

With portability, convenience and aesthetic design becoming increasingly important to consumers, the demand for flexible electronic device technology is also growing [1]. One of the most important properties of flexible electronic devices is their stable mechanical properties under repeated folding or bending [2]. Flexible transparent substrate is a vital part of flexible electronic equipment. Usually, transparent plastic, such as Polyethylene terephthalate (PET), Acrylonitrile Butadiene Styrene copolymer (ABS copolymer) and Polyvinyl chloride (PVC), etc., are used as substrate for flexible electronic devices because of their good mechanical properties, light weight, and low cost [3]. However, in recent years, the environmental problems caused by non-renewable and non-biodegradable resources are on the increase. As petroleum follow-on products, these transparent plastic products are neither biodegradable nor renewable, and their non-sustainability and non- environmentally friendly defects limit their application in the future [4]. Consequently, the demand for bio-based materials is urgent, and more attention is being paid to the sustainable production of biofuels, chemicals, and materials from lignocellulosic biomass [5,6,7,8].

Wood is the most abundant bio-based resource on the earth. It has been widely used for construction structural materials and decorative materials due to its high ratio of strength to weight, aesthetics, low thermal conductivity, and biodegradability [9,10,11]. As a major component of wood, cellulose plays an important role on the contribution of the mechanical properties of wood, which is usually used as a reinforcement for bio-based composite materials [12,13]. In recent years, nanocellulose, which is isolated from cellulose by a series of physical and chemical treatments, has received extensive attention due to its extremely high aspect ratios, excellent mechanical behavior, biodegradability and so on [6,14,15,16,17]. Among the substrates for flexible electronic equipment, nanocellulose composites are expected to replace glass or plastics in displays, optoelectronics or automobiles because of their high transparency, excellent mechanical properties, safety, and sustainability [18]. However, during the fabrication of nanocellulose, the isolation process of cellulose always consumes a lot of energy, moreover, the intensive chemical treatment destroys the hierarchical structural of the cell wall, which leads to the oriented nanocellulose architecture no longer existing [10,19].

In order to reduce the expenditure of energy and retain the original oriented nanocellulose structure of wood, the bulk wood is directly subjected to delignification treatment to produce hierarchical micro and nano-scale cellulose fibers with a diameter less than the wavelength of visible light [20]. Subsequently, the delignified wood is impregnated by polymers whose refractive index match cellulose, such as epoxy resin and polymethyl methacrylate [3,4,21,22]. The fabricated transparent wood not only retains the natural wood structure, but also possesses the advantages of nanocellulose.

Compared to natural wood, transparent wood has high transparency and toughness due to delignification and the penetration of transparent polymer [10,23]. At present, basswood [19], birch [22,24], and balsa [9] have already been successfully used to fabricate transparent wood. This transparent wood displays outstanding transparency, but the tensile strength is not satisfied because transversal wood is used as the substrate. Additionally, the dimension of these substrates is limited by the diameter of the trees, which is not suitable for a commercial process.

Fast-growing poplar is a type of artificially planted wood, which grows fast and whose growth cycle is around 9–12 years [25]. As a fast-growing raw material, the density is low and its mechanical properties are not very good, so usually, it is used to produce ordinary plywood for decoration or low-grade furniture with low economic value added. Recently, the quarter-sawed lumber of fast- growing poplar has also been used to prepare transparent wood in a few research studies [7], similarly, this kind of substrate has a low utilization rate and it is difficult to process on a large scale.

At present, the rotary cut veneer is the main processing unit material for fast-growing poplar, which has higher outturn percentage, and its dimensions and thickness are easily acquired according to need. Therefore, in this research, the rotary cut poplar veneer was used as the substrate to fabricate flexible transparent poplar veneer. Lignin was gradually removed from PV using acid sodium chlorite solution by controlling the delignification time, and epoxy resin was used as the filling in the delignified PV under vacuum impregnation, the preparation diagram is shown in Figure 1. Although epoxy resin is an unsustainable material, here it was chosen to fill PV because its refractive index matches cellulose. The study mainly concerns the effect of lignin content on microstructure, light transmittance, haze, mechanical strength, and thermal stability of the poplar veneer impregnated with epoxy resin.

## 2. Materials and Methods

### 2.1. Materials and Chemicals

Rotary cut poplar veneer was obtained from Siyang County, Jiangsu Province (density: 0.429 g/cm^3^, thickness: 1.0 mm), sodium chlorite (NaClO_2_, 80 % solid, Shanghai McLean Biochemical Technology Co., Ltd., Shanghai, China), Acetic acid (CH_3_COOH, AR, Nanjing Chemical Reagent Co., Ltd., Nanjing, China), ethanol (C_2_H_6_O, AR, Sinopharm Chemical Reagent Co., Ltd., Shanghai, China), epoxy resin (AB style, JH-5511 Hangzhou Wuhuigang Adhesive Co., Ltd. Hangzhou, China). Deionized water was prepared using the laboratory ultrapure water machine (VE-A-10 L/h, Hongseng Co., Ltd., Nanjing, China).

### 2.2. Fabrication of Flexible Transparent Poplar Veneer

#### 2.2.1. Delignification Process

Rotary cut poplar veneer (PV) with a size of 50 mm (Length) × 50 mm (Width) × 1.0 mm (Thickness) was dried at 60 °C for 24 h using a vacuum drying oven. The NaClO_2_ solution (5 wt%) with a pH of 4.6 was prepared to remove the lignin from the PV. Then, the PV was wholly immersed into the NaClO_2_ solution using a Teflon standoff as a shield, in which the mass ratio of the PV to the NaClO_2_ solution was 1:60. Then the delignification reaction was continued with an agitator at 75 °C. The PV was taken out after 1 h and washed with deionized water three times. Then the washed PV was placed into fresh NaClO_2_ solution to repeat the above reaction process. In order to obtain different lignin contents, the PV was treated for 2 h, 4 h, 6 h, 8 h, and 10 h, respectively, separately named as PV_2_, PV_4_, PV_6_, PV_8_, and PV_10_. Subsequently, all delignified PVs were purged with a mixed solution of ethanol:deionized water to remove the remaining chemical substances. Finally, the cleaned PVs were freeze-dried to be further used.

#### 2.2.2. Fabrication of Flexible Transparent PV

The curing agent was added to the epoxy resin at a mass ratio of 1:3, and after stirring evenly, the delignified PV was immersed in the epoxy resin under vacuum pressure for 5 min, then, the delignified PV was impregnated for 10 min under atmospheric pressure. In order to ensure sufficient penetration, the above impregnation process was repeated three times. Subsequently, the excess epoxy resin was removed from the surface of the PV using a razor blade, and then the PV was polymerized at 60 °C for 2 h to obtain different PV specimens, named as TPV_2_, TPV_4_, TPV_6_, TPV_8_, and TPV_10_, respectively.

### 2.3. Characterization

The microstructure of the cross-section of all delignified PVs and TPVs was observed using an environmental scanning electron microscope (ESEM) (Quanta 200, FEI, Hillsborough, Oregon, USA). Before the testing, all specimens were dried at 60 °C for 12 h, and then the observation surfaces of all specimens were sprayed with gold using a particle sputtering instrument. The testing voltage was 15 kV and the analysis mode resolution was 3.0 nm

FTIR spectroscopy was performed on a spectrometer (FTIR VERTEX 80V, Bruker, Karlsruhe, Germany) at room temperature, all delignified PV specimens were dried at 80 °C in a vacuum-dryer for 12 h before detection. Then a small quantity of each specimen was blended with KBr powder and compressed to form a disk. The investigated range was 400–4000 cm^−1^ at a resolution of 4 cm^−1^.

The Van Soest method (GB/T 20806-2006 Determination of Neutral Detergent Fiber (NDF) in Feed) was used to determine the relative content of cellulose, hemicellulose, and lignin of all the delignified PVs.

The X-ray diffraction (XRD) analysis of all delignified PVs was performed by X-ray diffractometer (Ultima IV, Rigaku, Tokyo, Japan) using Ni-filtered Cu Ka radiation (λ = 1.5406 Å) at 40 kV and 30 mA. Scattered radiation was recorded in the range of 5–50° at a scan rate of 5°/min.

According to ASTM D1003 “Standard Method for Haze and Light Transmittance of Transparent Plastics”, the light transmittance and haze of all the TPVs were detected using a UV–Vis spectrophotometer (Lambda 950, PE, Walsham, MA, USA) in the spectrum range of 200–800 nm.

The tensile strength of all TPVs with a dimension of 50 mm (L) × 5 mm (W) × 1 mm (H) was measured using a microcomputer-controlled electronic universal testing machine (SANS 4304, MTS, San Diego, CA, USA). The tensile rate was 5 mm/min and each group had at least six specimens of which the averages were obtained.

The thermal stability of all TPVs was investigated with a thermogravimetric analysis instrument (STA8000, Perkin Elemer, Waltham, MA, USA). Approximately 8 mg of specimen was used for each test and every specimen was repeated at least three times. The testing temperature range was 30–700 °C at a heating rate of 10°/min under a nitrogen atmosphere.

## 3. Results and Discussion

### 3.1. The EFFECT of Delignification Time on the Microstructure and Chemical Composition of PV

#### 3.1.1. The Effect of Delignification Time on the Microstructure of PVs

Natural wood is composed mainly of cellulose, hemicellulose, and lignin, of which cellulose and hemicellulose are colorless in the visible light range, while lignin plays an important role for the color of wood due to the strong absorption of visible light. Generally, the brown and opaque wood is mainly attributed to the strong absorption of visible light of the lignin and light scattering from its uneven structure [26]. In order to prepare transparent wood, it is necessary to remove lignin to reduce the light attenuation caused by light absorption [9,27]. In the study, the effect of delignification could be first judged by the color changes of the PV [28]. As shown in Figure 2, it can be clearly seen that as the delignification time increased, the color of the PVs gradually changed from chalky yellow to white. After 8~10 h of delignification, The PV almost became plain white, and it was speculated that the lignin had been removed completely.

Figure 2 shows the microstructure of the cross-section of PV and all delignified PVs with different reaction times. The cells of PV were arranged densely, in which the cell wall was intact and the middle lamella between the wood cells was not destroyed. However, after delignification for 2 h and 4 h, part of the middle lamella was destroyed, and obvious gaps appeared in the cross-section of the PV due to the partial removal of lignin.

Having been delignified for 6 h, the middle lamella was destroyed and the cell wall was slightly thinned, also many gaps appeared in the cross-section of the PV. These facts indicated that most of the lignin had been removed. The middle lamella had almost completely disappeared in PV_8_, and the transect of the cell wall showed a clear layered structure, the intercellular space and cell cavity were obviously enlarged, and the porosity increased significantly. These phenomena indicated that the lignin in the cell wall and middle lamella were probably removed completely after delignification for 8 h.

When the delignification time increased to 10 h, the intercellular space and cell cavity of the PV were further enlarged, however, the cell wall significantly contracted and collapsed. The phenomenon could be ascribed to the damage of the cellulose crystal structure due to a long delignification time.

#### 3.1.2. The Effect of Delignification Time on the Lignin Content of PV

Figure 3 shows the FTIR spectra of PV and all delignified PVs. On the FTIR spectrum curve of PV, the characteristic peaks located at 1505 cm^−1^, 1425 cm^−1,^ and 1235 cm^−1^ represent the aromatic skeleton vibrations and the benzene ring-hydrogen bond stretching vibration of lignin, respectively. The characteristic peaks located at 1462 cm^−1^ and 1367 cm^−1^ are ascribed to the C–H bending vibration absorption of lignin. The peak located at 1631 cm^−1^ is due to the conjugated carbonyl (C=O) stretching vibration of lignin. However, with the increase of the delignification time, the intensity of these characteristic peaks was gradually weakened—of note, these absorption peaks could scarcely be found on the FTIR spectrum curves of PV_8_ and PV_10_. This fact indicated that the lignin of the PV was removed completely after 8 h.

On the other hand, the intensity and position of the characteristic peaks located at 3430 cm^−1^, 2913 cm^−1^, and 895 cm^−1^ were nearly unchanged for all FTIR spectra. These peaks are mainly ascribed to the O–H and C–H stretching vibration of cellulose. The characteristic peak of hemicellulose located at 1730 cm^−1^, which is caused by the C=O stretching vibration, was not changed except for the intensity which was slightly decreased after delignification for 8 h. These facts revealed that the cellulose and hemicellulose of the PV were seldom affected by the delignification process.

The relative content of cellulose, hemicellulose and lignin of all PVs was quantitatively measured by the Van Soest method [29]. The results are shown in Table 1. The lignin decreased sharply from 22.69 to 8.31% after delignification for 2 h, and the lignin content gradually reduced with the increase of delignification time. The relative content of lignin was about 0.41% after 8 h, which indicated that the lignin had been almost removed completely. When the delignification time was at 10 h, the relative content of lignin decreased to 0.06%. Correspondingly, the relative content of cellulose and hemicellulose gradually increased with the increase of delignification time. Finally, the relative content of cellulose and hemicellulose increased to 78.89% and 20.66%, respectively, when the delignification time was 8 h. However, the relative content of cellulose slightly decreased after 10 h of delignification.

#### 3.1.3. Effect of Delignification Time on the Cellulose Crystal Structure of PV

As shown in Figure 4, the diffraction peaks appearing at 2θ = 15.8° and 22.6° are due to the *I*_101_ and *I*_002_ crystal planes of cellulose, and a small diffraction peak appearing at 2θ = 35° is the *I*_040_ crystal plane of cellulose [7]. These characteristic peaks represent a typical cellulose I crystal structure. The diffraction curves of all PVs displayed the same diffraction peaks, which indicated that the delignification process did not change the cellulose crystal structure. On the other hand, the intensity of the *I*_002_ diffraction peak gradually increased with the increase of the delignification time, nevertheless, it significantly decreased for PV_10_. According to the Segal method, the cellulose crystallinity of all PVs was calculated, as shown in Table 2. The crystallinity of PV was 53.8%, which increased to 69.8% for PV_8_, while the cellulose crystallinity decreased to 62.7% for PV_10_. These facts showed that the removal of lignin increased the crystallinity of cellulose, but excessive delignification treatment decreased the crystallinity of cellulose.

### 3.2. Microstructure and Properties of Flexible Transparent Poplar Veneer (TPV)

#### 3.2.1. Microstructure of TPV

The main reason for opaque wood is due to the existence of lignin and the porosity. In order to fabricate transparent wood, therefore, the lignin needs to be removed and the interspaces in the wood need to be filled with polymers, whose refractive index match cellulose (1.53). In this study, epoxy resin, whose refractive index is about 1.5, was used to fill the interspaces and cell cavities of the PVs with the different lignin contents [9]. The physical object of the different TPVs is shown in Figure 5. It is clearly seen that the TPV_2_ is yellow and semitransparent, however, the TPVs gradually became colorless and transparent with the decrease of lignin content, finally, the TPV_8_ and TPV_10_ have almost complete achromaticity and transparency, which corresponds to the lignin content of the PVs listed in Table 1.

The microstructure of the cross-section of all TPVs is also displayed in Figure 5. It clearly indicates that the epoxy resin was filled into the vessels and cell cavities of all PVs, moreover, the impregnation process did not destroy the framework of the cell wall of the PV [3]. Nevertheless, PV_2_ and PV_4_ did not provide a large amount of void space due to the high lignin content, and the epoxy resin did not form a complete filling in the wood cells. However, the filling of epoxy resin in PV_6_ became full and dense, as just a handful of intercellular spaces could be found. When the lignin was entirely removed from the cell wall, as shown in TPV_8_ and TPV_10_, the epoxy resin was completely filled in the cell wall and cavity, and there was no interface detachment among the intercellular space and epoxy resin. Notably, the interface could practically not be observed between the cell wall and the polymer for TPV_10_. The void ratio of the PVs gradually increased with the removal of lignin and the moving channel of the epoxy resin increased. Subsequently, the epoxy resin enters the cell wall to replace the location of the lignin through microcapillary, which makes the microscopic voids decrease. Usually, the refractive index of air is about 1.0, which is mismatched with light. Therefore, the decrease of micropores has a great influence on the light transmittance of the TPVs [30].

#### 3.2.2. Optical Properties of TPVs

The optical properties (mainly involving transparency and haze) of the substrates play a crucial role in the preparation of optoelectronic devices [31]. In current reports on glass, plastic and nano- paper substrates, the maximum transparency is about 90%, but the optical haze is less than 20% [32]. Optical haze is defined as the ability to scatter incident light, appearing as a translucent or opaque state [31]. A higher optical haze means a better interaction between the light and the active medium, which can improve the efficiency of the optical device [9]. Transparent wood with high light scattering has potential applications in solar cells, OLED lighting systems, liquid crystal display (LCD) backlight units, signage, etc. [33].

The transmittance and haze of all TPVs in the spectral range of 200–800 nm is shown in Figure 6. As shown in Figure 6a,b the light transmittance of TPV_2_ was only 45% due to the high lignin content and the incomplete filling of epoxy resin, which increased the scattered light. With the further removal of lignin from the cell wall and intercellular layer, the porosity gradually increased, which allowed the epoxy resin to be fully permeated into the PV. Therefore, the light transmittance of TPV_4_ and TPV_6_ was greatly enhanced, whose values achieved 68% and 77%, respectively. When the lignin was completely removed, TPV_8_ and TPV_10_ reached high light transmittance of 81% and 82% respectively, which are near to the light transmittance of pure epoxy resin of around 88%. Figure 6c shows the haze of all TPVs in the range of 400 to 800 nm, in which TPV_8_ displayed the highest haze value of 62% at 400 nm, while the haze value of TPV_10_ was about 61%, which was slightly lower than that of TVP_8_. Additionally, TPV_4_ shows the lowest haze value at 400 nm. Notably, the haze values of all TPVs decreased slightly with the increase of the wavelength beyond 473 nm, and the haze values decreased to 46–53% at 800 nm. In addition to the outstanding optical properties, the TVP should be flexible enough to be used in electronic devices. In this study, the flexibility of TPV_8_ was evaluated due to its relatively high light transmittance and haze values of all the TPVs, as shown in Figure 6d–f. The findings displayed that the lowest bending radius of TPV_8_ could reach 2.5 mm and it could be tightly enlaced into a roll without transverse cracks, also, the original shape of TPV_8_ could be rapidly restored after the force was released. In addition, it still maintained good optical properties under the bending state. Therefore, the fabricated TPV_8_ in this study has enormous potentials in photovoltaic applications, such as photovoltaics, solar cells, smart windows, and light shaping diffusers [7,34].

#### 3.2.3. Thermal Stability of the TVPs

The thermogravimetric curves (TG) and the differential thermal curves (DTG) of all TPVs and pure epoxy resin are shown in Figure 7. The thermal degradation of all TPVs mainly included four stages [35]. The first stage from room temperature to 120 °C, is the dehydration drying of the TPVs, which is mainly due to the evaporation of the free moisture and absorbed water of TPVs, and the TG curves in this range were relatively flat due to little weight loss. The second stage occurred from 150 °C to 230 °C, which is mainly due to the pyrolysis of hemicellulose and partially the lignin of the TPVs, also including a small quantity of epoxy resin [36]. All TPVs exhibited a similar DTG curve in this stage and the weight loss rate was about 7%. The third stage is between 230 °C and 350 °C, which is mainly ascribed to the pyrolysis of the remaining lignin and partially pyrolysis of epoxy resin [37,38]. Because of the high lignin content, the pyrolysis rate of TPV_2_ and TPV_4_ increased rapidly, and their weight loss reached 24% and 18%, respectively. However, the TPV_8_ displayed the lowest weight loss and pyrolysis rate, which further indicated that the lignin had been removed completely. The fourth stage is from 350 to 450 °C, which is ascribed to the pyrolysis of cellulose, residual lignin, and the remaining epoxy resin [39]. The weight loss rate of TPV_8_ was at 64% at this stage, the weight loss rates of TPV_2_ and TPV_4_ were 48% and 54%, respectively. Additionally, the DTG curves showed significantly that the pyrolysis rate of TPV_2_ and TPV_4_ was higher than that of TPV_8_, the decomposition rate of epoxy resin was significantly higher than that of the TPVs. It is the carbonization stage of the TPVs above 450 °C, where the thermal decomposition was basically finished and the TG and DTG curves were relatively gentle. Thermogravimetric analysis further proved the thermal stability of TPV was improved due to the removal of lignin, which increased the penetration of epoxy resin and delayed the pyrolysis of PV [40]. Notably, the thermal stability of TPV_10_ is lower than that of TPV_8_, which is possibly ascribed to the decrease of cellulose crystallinity.

#### 3.2.4. Density and Tensile Strength of TPVs

The density of all PVs and TPVs is displayed in Figure 8a. Obviously, the density of the PVs gradually decreased with the increase of delignification time. After 10 h of delignification, the density of PV decreased to 0.279 g/cm^3^ due to the complete removal of lignin. On the other hand, the density of TPVs was significantly enhanced because of the filling of epoxy resin. It reached a maximum density of 1.241 g/cm^3^ for TPV_8_, and the filling amount of epoxy resin was 237% of the mass of PV_8_. This is because the microchannels and voids of PV gradually increased with the removal of lignin, which correspondingly augmented the filling of epoxy resin. The tensile strength of TPVs was measured along the longitudinal direction of the PV, as shown in Figure 8b. The tensile strength of TPVs was greatly reduced with the reduction of lignin content. The tensile strength of PV was 114 MPa, and the tensile strength of TPV_2_, TPV_4_, TPV_6_, TPV_8_, TPV_10_ decreased to 97 MPa, 88 MPa, 81 MPa, 67 MPa, and 54 MPa, respectively. In wood, cellulose existing in the cell wall is the skeleton material of the cell wall, and lignin permeates into the cellulose skeleton as a binder to strengthen the cell wall [41,42]. With the removal of lignin, the location of lignin was occupied by epoxy resin, and the voids of the cell wall were also filled. However, the epoxy resin could not play a role as binder among the microfibrils, and although the epoxy resin could react with the alcohol group of cellulose, the main interaction between cellulose and epoxy resin was physical contact. Moreover, the pure epoxy resin used in this study has a low tensile strength, which is about 1 MPa. Therefore, the values of tensile strength of TPVs significantly reduced with the increase of impregnation of epoxy resin. All the same, the tensile strength of TPV_8_ still reached 67 MPa, which is significantly higher than the values of the transparent plastic substrates used in flexible electronic devices (such as polyethylene terephthalate, whose the tensile strength is around 50 MPa [43]).

## 4. Conclusions

In this study, in order to fabricate flexible transparent PV, the rotary cut veneer from fast- growing poplar was first delignified followed by filling with epoxy resin. The research indicated that the cell wall became thin and the porosity of PV was significantly augmented with the removal of lignin, while the lignin was almost completely removed from the cell wall after delignification for 8 h. Meanwhile, the cellulose crystallinity of PV was greatly increased with the decrease of lignin content. SEM observation indicated that the epoxy resin had been filled into the cell cavities and cell wall of the TPVs, and the filling amount of epoxy resin gradually increased with the decrease of lignin content. Correspondingly, the density, light transmittance, and haze of all TPVs distinctly increased, and the thermal stability became more stable; however, the tensile strength of all TPVs significantly declined. Among them, TPV_8_ exhibited excellent optical properties and thermal stability, of which the light transmittance and haze achieved 82% and 62%, respectively. Also, the tensile strength of TPV_8_ still reached 67 MPa, which is significantly higher than that of traditional plastic substrates used in flexible electronic devices. Although TPV_10_ has excellent optical properties, its tensile strength and thermal stability were lower than those of TPV_8_. Moreover, TPV_8_ still displayed fine optical properties under bending state, and the minimum bending radius achieved 2.5 mm, which displayed outstanding flexibility. Therefore, as a type of wood-based flexible transparent material, TPV_8_ displayed great potential to substitute for plastics as a substrate to be used in solar cells, OLED lighting systems, liquid crystal display (LCD) backlight units, signage, etc.

## Figures and Tables

**Figure 1 polymers-12-02602-f001:**
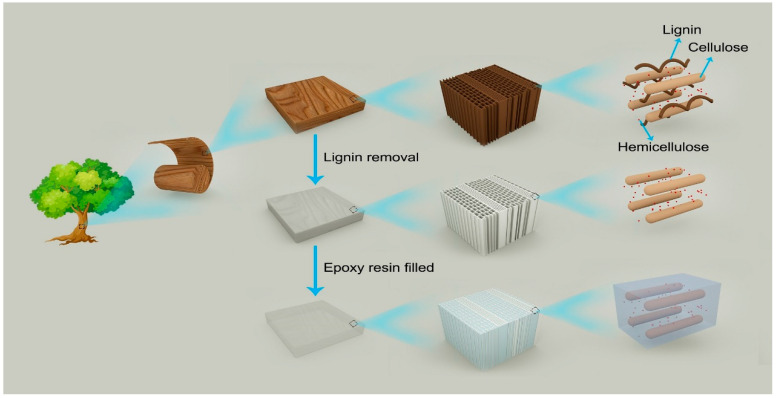
Schematic diagram of TPVs prepared from PV.

**Figure 2 polymers-12-02602-f002:**
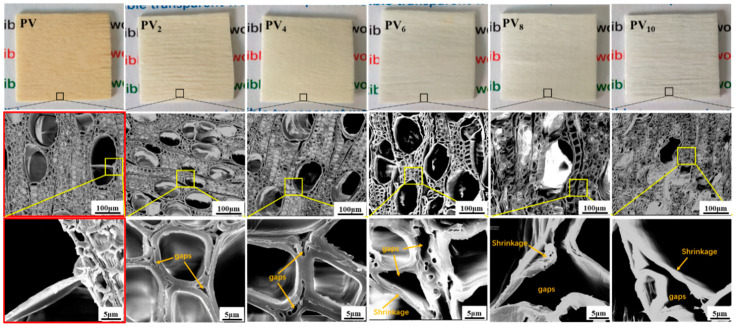
The physical objects and the microstructure of the cross-section of PV, PV_2_, PV_4_, PV_6_, PV_8_, and PV_10_ under 300 and 5000 magnifications, respectively.

**Figure 3 polymers-12-02602-f003:**
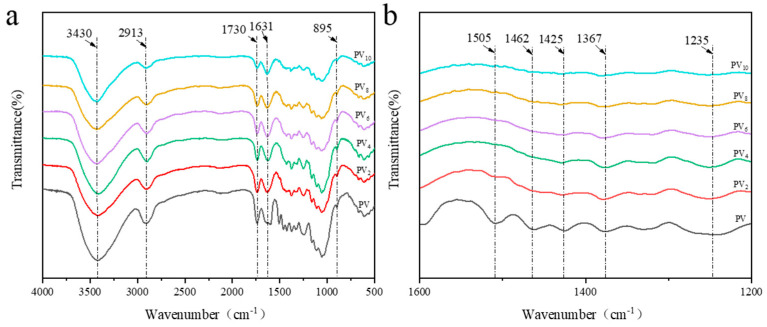
FTIR spectra of PV and PVs with different delignification time. (**a**) FTIR spectrum of the 4000–500 band, (**b**) FTIR spectrum of the 1600–1200 band.

**Figure 4 polymers-12-02602-f004:**
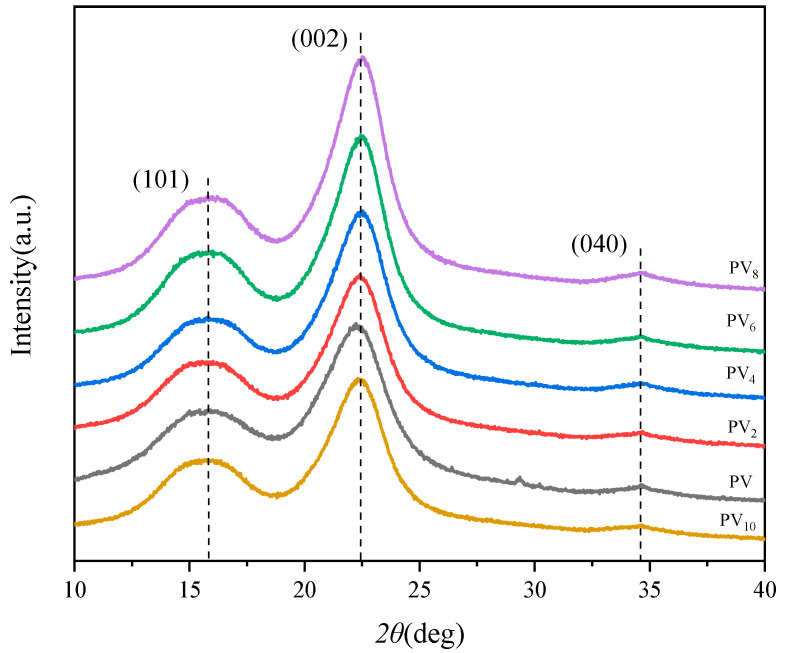
XRD patterns of PVs with different delignification time.

**Figure 5 polymers-12-02602-f005:**
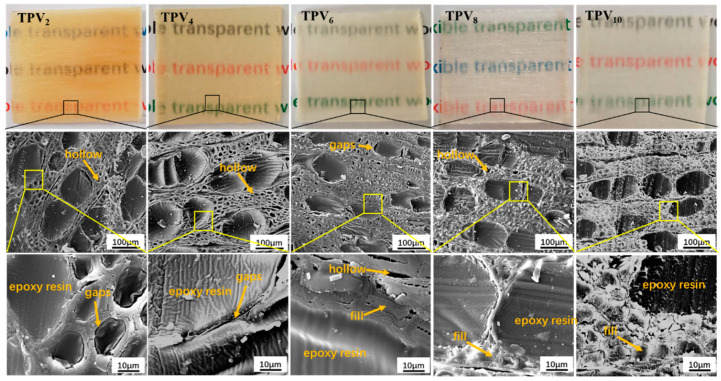
The physical object and the microstructure of TPV_2_, TPV_4_, TPV_6_, TPV_8_ and TPV_10_ under a magnification of 300 and 2500, respectively.

**Figure 6 polymers-12-02602-f006:**
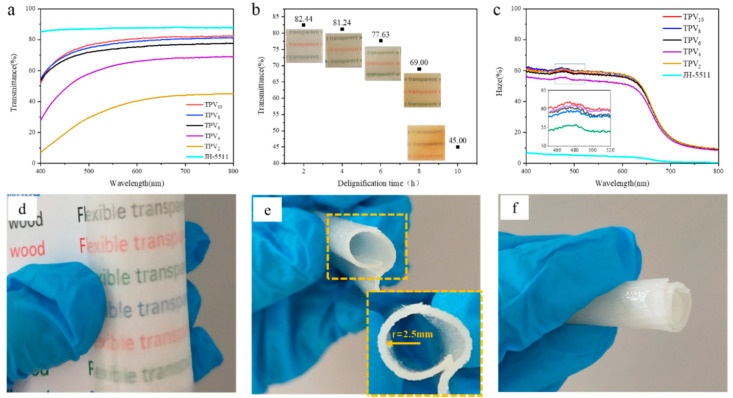
Optical properties and flexibility of TPVs. (**a**) Transmittance curves, (**b**) the actual light transmittance comparison chart at 800 nm, (**c**) haze curves, (**d**) optical performance of TPV_8_ under bending, (**e**) the bending radius of TPV_8_, (**f**) curled diagram of TPV_8_.

**Figure 7 polymers-12-02602-f007:**
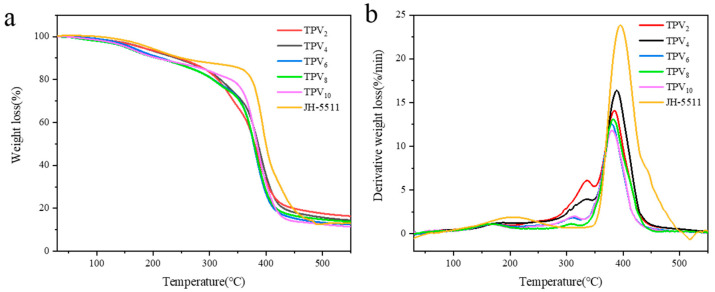
TG and DTG curves of TPVs. (**a**) TG curves, (**b**) DTG curves.

**Figure 8 polymers-12-02602-f008:**
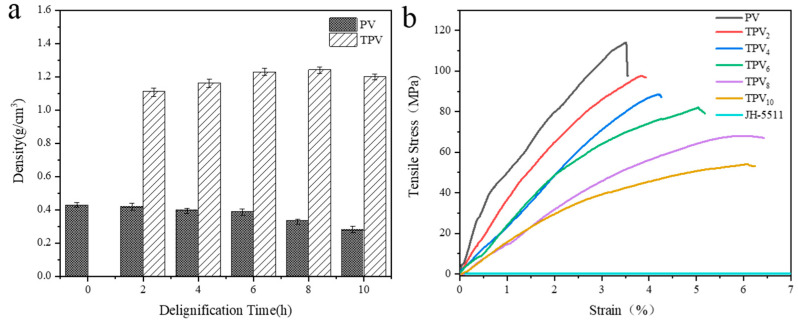
Density and stress-strain curves of PV and all TPVs. (**a**) Density diagram, (**b**) Stress strain curve.

**Table 1 polymers-12-02602-t001:** The relative content of cellulose, hemicellulose, and lignin of all PVs with different delignification time.

Sample	Hemicellulose (%)	Cellulose (%)	Lignin (%)
PV	15.30	61.80	22.69
PV_2_	21.62	70.06	8.31
PV_4_	23.19	73.28	3.51
PV_6_	21.58	75.66	2.71
PV_8_	20.66	78.89	0.41
PV_10_	21.34	78.56	0.06

**Table 2 polymers-12-02602-t002:** The relative crystallinity of cellulose for all PVs with different delignification time.

Sample	Relative Crystallinity (%)
PV	53.8
PV_2_	61.4
PV_4_	65.3
PV_6_	67.2
PV_8_	69.8
PV_10_	62.7

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
