# Peer review of "Effect of Lignin Content on Properties of Flexible Transparent Poplar Veneer Fabricated by Impregnation with Epoxy Resin"

_polymers, 2020, doi:10.3390/polym12112602_

Round 1

Reviewer 1 Report

The manuscript "Effect of lignin content on properties of the flexible transparent poplar veneer fabricated by impregnation of epoxy resin" by He and coworkers deals with the formation of cellulose-based materials. Overall, the manuscript covers an interesting topic. Nevertheless, the manuscript needs extensive english editing to be publishable.

General comments:

1.Is there any option to use the released lignin? Is it only washed out or degraded as well?

2.Although the authors use wood, the used resin is not sustainable, which should be discussed. How much resin is present in the final material?

3.Why does the removal of lignin increase the crystallinity of cellulose?

4.What is the interaction of resin and cellulose? Is it chemical or physical?

5.The authors should also show tensile measurement and TGA of pure resin.

6.The authors mention degradable materials in the conclusion, but with the resin the material will not be degradable.

7.Why is there a maximum in haze for the 8 h sample?

8. What is the effect of the NaClO2 solution concentration or the NaClO2 amount? What is the effect of reaction temperature?

Specific comments:

9.The reference section has to be revised. Journal abbreviations, authors names, journal names checked.

10.The abstract should be in present tense

11.The title should be "... with epoxy resin"

12.The authors use "remove" very often, where "removal" would be more appropriate.

13.ABS should be "ABS copolymer"

14.On page 2 line 40, the sentence does not make sense: the safety of the supply for industry is not increasing

15.On page 4, cm-1 needs a superscript

16.The units should be written with a space: xx unit

17.The authors should avoid to use couldn't.

18.What does the word "effection" mean?

19. "react" should be "reaction"? "Ph" should be "pH"

20.There are several "Error! Reference...." in the manuscript

Author Response

General comments:

1.Is there any option to use the released lignin? Is it only washed out or degraded as well?

Reply:Lignin is a kind of aromatic compound, which can be used as an important chemical material to produce vanillic aldehyde, rubber reinforcing filler, and so on. However, in this research, the lignin was completely degraded by NaClO2 solution (5 wt. %) with a pH of 4.6, it is difficult to obtain pure lignin.

2.Although the authors use wood, the used resin is not sustainable, which should be discussed. How much resin is present in the final material?

Reply:In this research, the amount of epoxy resin gradually increased with the decrease of lignin, the maximum was about 50%. Just like you said, the manufactured transparent wood is not sustainable, we will supplement discuss in the introduction.

3.Why does the removal of lignin increase the crystallinity of cellulose?

Reply:There are two probable causes to increase the crystallinity of cellulose due to the removal of lignin. The first one was that the ether bond between lignin and cellulose was broken, which caused the recrystallization of the exposed hydroxyl groups in the cellulose molecules. The second is because the hydroxyl groups existing amorphous crystalline region was activated due to the hydro-thermal reaction, then formed hydrogen bond, which enhanced the cellulose molecules arrange in the amorphous crystalline region.

4.What is the interaction of resin and cellulose? Is it chemical or physical?

Reply:it is physical interaction between epoxy resin and cellulose.

5.The authors should also show tensile measurement and TGA of pure resin.

Reply:we already added the tensile strength and TGA of pure resin in the figure and text, please check it.

6.The authors mention degradable materials in the conclusion, but with the resin the material will not be degradable.

Reply:we allready modified the conclusion, please check it.

7.Why is there a maximum in haze for the 8 h sample?

Reply:Haze is mainly produced by light scattering of microcelluloses of wood, the haze of wood increased with the remove of lignin, which increase the amount of microcelluloses and  porosity, however, when the treatment time reached 10 h, the crystallinity of cellulose decreased and the microcelluloses and porosity were brokened, which caused the reduce of haze.

  1. What is the effect of the NaClO2 solution concentration or the NaClO2 amount? What is the effect of reaction temperature?

Reply:The higher NaClO2 solution concentration is, the more reactive ion, which will completely break the structure of wood. Therefore, we need control the ration of NaClO2 and wood mass. Similarly, the higher reaction temperature will break the structure of wood.

Specific comments:

9.The reference section has to be revised. Journal abbreviations, authors names, journal names checked.

Reply:we have made the revise according to your suggestion.

10.The abstract should be in present tense

Reply:we have made the revise according to your suggestion.

11.The title should be "... with epoxy resin"

Reply:we have made the revise according to your suggestion.

12.The authors use "remove" very often, where "removal" would be more appropriate.

Reply:we have made the revise according to your suggestion.

13.ABS should be "ABS copolymer"

Reply:we have made the revise according to your suggestion.

14.On page 2 line 40, the sentence does not make sense: the safety of the supply for industry is not increasing

Reply:we have made the revise according to your suggestion.

15.On page 4, cm-1 needs a superscript

Reply:we have made the revise according to your suggestion.

16.The units should be written with a space: xx unit

Reply:we have made the revise according to your suggestion.

17.The authors should avoid to use couldn't.

Reply:we have made the revise according to your suggestion.

18.What does the word "effection" mean?

Reply:we have made the revise according to your suggestion, replace effection with effect.

  1. "react" should be "reaction"? "Ph" should be "pH"

Reply:we have made the revise according to your suggestion.

20.There are several "Error! Reference...." in the manuscript

Reply:we have made the revise according to your suggestion.

Reviewer 2 Report

Article is well written . There are some corrections to be done as also suggested in the attached file:

  • leave a space between text and number of reference;
  •  leave space between number and units of measurement;
  • you write 15 KV and 40 Kv - in both ways it is wrongly written; unit is kV
  • place all figures in proper location; it is very difficult to follow them;
  • in many places, when citing references appears Error! Reference source not found.; make everywhere corrections.

Author Response

  1. leave a space between text and number of reference;

Reply:we have made the revise according to your suggestion.

  1. leave space between number and units of measurement;

Reply:we have made the revise according to your suggestion.

  1. you write 15 KV and 40 Kv - in both ways it is wrongly written; unit is kV

Reply:we have made the revise according to your suggestion.

  1. place all figures in proper location; it is very difficult to follow them;

Reply:we have made the revise according to your suggestion.

5.in many places, when citing references appears Error! Reference source not found.; make everywhere corrections.

Reply:we have made the revise according to your suggestion.

Round 2

Reviewer 1 Report

The revised version of the manuscript is improved. Nevertheless, I still have some comments:

1. I was wondering if the interaction between resin and cellulose is chemical because epoxy can also react with alcohols like in cellulose?

2. I could not find the discussion about sustainability of epoxy in the introduction.

3. Why is the strain curve of PV and not TPV shown (maybe its an error)?

4. The manuscript still contains the "Error...."

5. The authors should avoid "didn't"

Author Response

  1. I was wondering if the interaction between resin and cellulose is chemical because epoxy can also react with alcohols like in cellulose?

Reply: In this study, we thought that the connection between resin and cellulose was mainly physical interaction, of course, as you judged, the epoxy may react with alcohols in cellulose, however, we think it is not majority. Also, we will add this reason to the related content.

  1. I could not find the discussion about sustainability of epoxy in the introduction.

Reply: Actually, in this research, we mainly use the sustainable wood veneer as the mainly substrate to fabricate transparent wood veneer, while epoxy resin is not sustainable, we will explain it in the introduction.

  1. Why is the strain curve of PV and not TPV shown (maybe its an error)?

Reply: we already modified it.

  1. The manuscript still contains the "Error...."

 Reply: we already modified it.

  1. The authors should avoid "didn't"

Reply: we already modified it.

This manuscript is a resubmission of an earlier submission. The following is a list of the peer review reports and author responses from that submission.